# Analysis of the Mechanism of Wood Vinegar and Butyrolactone Promoting Rapeseed Growth and Improving Low-Temperature Stress Resistance Based on Transcriptome and Metabolomics

**DOI:** 10.3390/ijms25179757

**Published:** 2024-09-09

**Authors:** Kunmiao Zhu, Jun Liu, Ang Lyu, Tao Luo, Xin Chen, Lijun Peng, Liyong Hu

**Affiliations:** 1Hubei Key Laboratory of Nutritional Quality and Safety of Agro-Products, Institute of Quality Standard and Testing Technology for Agro-Products, Hubei Academy of Agricultural Sciences, Wuhan 430072, China; zhukunmiao@hbaas.com (K.Z.); liujun@hbaas.com (J.L.); lvang2007@foxmail.com (A.L.); hgchenxin@163.com (X.C.); 2Ministry of Agriculture Key Laboratory of Crop Ecophysiology and Farming System in the Middle Reaches of the Yangtze River, College of Plant Science and Technology, Huazhong Agricultural University, Wuhan 430070, China; luotao28@webmail.hzau.edu.cn

**Keywords:** rapeseed, wood vinegar, butyrolactone, transcriptome and metabolomics, yield, low-temperature resistance

## Abstract

Rapeseed is an important oil crop in the world. Wood vinegar could increase the yield and abiotic resistance of rapeseed. However, little is known about the underlying mechanisms of wood vinegar or its valid chemical components on rapeseed. In the present study, wood vinegar and butyrolactone (γ-Butyrolactone, one of the main components of wood vinegar) were applied to rapeseed at the seedling stage, and the molecular mechanisms of wood vinegar that affect rapeseed were studied by combining transcriptome and metabolomic analyses. The results show that applying wood vinegar and butyrolactone increases the biomass of rapeseed by increasing the leaf area and the number of pods per plant, and enhances the tolerance of rapeseed under low temperature by reducing membrane lipid oxidation and improving the content of chlorophyll, proline, soluble sugar, and antioxidant enzymes. Compared to the control, 681 and 700 differentially expressed genes were in the transcriptional group treated with wood vinegar and butyrolactone, respectively, and 76 and 90 differentially expressed metabolites were in the metabolic group. The combination of transcriptome and metabolomic analyses revealed the key gene-metabolic networks related to various pathways. Our research shows that after wood vinegar and butyrolactone treatment, the amino acid biosynthesis pathway of rapeseed may be involved in mediating the increase in rapeseed biomass, the proline metabolism pathway of wood vinegar treatment may be involved in mediating rapeseed’s resistance to low-temperature stress, and the sphingolipid metabolism pathway of butyrolactone treatment may be involved in mediating rapeseed’s resistance to low-temperature stress. It is suggested that the use of wood vinegar or butyrolactone are new approaches to increasing rapeseed yield and low-temperature resistance.

## 1. Introduction

As the third largest oil crop in the world, rapeseed is the most important source of vegetable oil in China [1]. However, compared with the high-yield countries in Europe, the yield per unit area of rapeseed in China is relatively low. Improving the seed yield and quality of oil crops has been the main research focus of the past decades. Rapeseed is the only winter oil crop in China. Low-temperature stress limits geographical distribution and seriously affects crop growth, development, yield, and quality [2,3]. Every year, crops worldwide lose hundreds of billions of yuan due to low-temperature damage. Therefore, in recent years, researchers have been committed to coordinating environmental and cultural measures to improve rapeseed yield and tolerance to low-temperature stress. Plant growth regulators are an important regulatory measure to achieve robust growth with a high and stable yield of rapeseed [4,5].

Wood vinegar is a semi-transparent reddish-brown liquid condensed from carbonized flue gas during biochar production from agricultural and forestry wastes under high-temperature and hypoxia conditions [6]. Wood vinegar contains various complex chemical components, as follows: organic acids, phenols, alkanes, furan derivatives, esters, alcohols, sugar derivatives, and nitrogen compounds [7]. As a natural source material of biomass, it has been widely used as a foliar fertilizer, insect repellent, and soil conditioner. In recent years, it has been found that wood vinegar is a compound plant growth regulator analog. Research shows that wood vinegar can accelerate the growth of roots, stems, and leaves by increasing the germination rate of seeds, thereby increasing the yield of rice, wheat, and tobacco [8,9,10]. Further research shows that wood vinegar can improve the tolerance of wheat to abiotic stress by increasing the activity of antioxidant enzymes and the content of soluble protein [11]. Therefore, wood vinegar has multiple effects, including promoting the healthy growth of crops, increasing yield, and enhancing stress resistance. However, research on the impact of wood vinegar on rapeseed has focused more on its morphology and physiology, but the related molecular regulation mechanisms are still unknown.

In our preliminary research, we found that wood vinegar contains at least 159 organic substances. Among the 54 substances with a relative content greater than 0.07%, we screened more than 10 substances that can significantly promote plant growth and improve resistance, belonging to acids, esters, aldehydes, phenols, alkanes, etc. Among them, gamma butyrolactone is the most effective ingredient. Butyrolactone is one of the main components of wood vinegar [12,13,14]. The basic core skeleton of many natural products in nature contain γ-Lactone rings; compounds containing a γ-Butyrolactone ring usually exhibit a wide range of biological activities, including participation in cell growth and antibacterial and insect-resistant properties, which makes them a special structure in new drugs of concern [15]. And that γ-Butyrolactones (GBLs) are used as hormone signals to regulate physiological behavior [16], activate the biosynthesis of various secondary metabolites, including antibiotics, and play an important role in participating in cell signal transduction and controlling cell development [17,18,19]. However, the study of butyrolactone on rapeseed has not been reported.

The latest technological advances in transcriptomics and metabolomics provide a new way to address changes in gene expression and metabolite accumulation related to crop yield and stress. A transcriptome analysis showed that genes involved in the cell wall and membrane stability and lipid metabolism, ABA biosynthesis, calcium signal transduction, the hormone, and soluble sugar and protein biosynthesis and metabolism pathways were significantly changed after cold stress [20,21,22]. A metabolomic analysis showed that sugar, lipids, and amino acids have been widely reported to participate in the cold-stress reaction, not only in the process of protein synthesis, but also as the precursor of other key metabolites in cold-stress reactions [23]. By integrating transcriptome and metabolome data, the co-regulation of genes and metabolites that regulate yield and low-temperature stress in many plants has been more widely understood [24]. For example, genes encoding proteins involved in the sucrose starch metabolism and glyoxylate cycle in rice are up-regulated. These changes are related to the accumulation of glucose, fructose, and sucrose in rice after cooling or dehydration [25]. Using 20 mutant lines in Arabidopsis thaliana, flavonoids are considered to be the determinant of frost resistance and cold acclimation [26]. Changes in arginine and proline metabolism in white grapeseed, figs, and purple jasmine are important in improving cold adaptation [27,28,29]. Recently, many studies on rapeseed have revealed gene pathways and metabolites related to increasing yield, plant height, lignin and fatty acid biosynthesis, cadmium stress, drought, and cold stress [30]. However, there are no reports on the molecular mechanisms of wood vinegar and butyrolactone on rapeseed yield and its tolerance to low-temperature stress.

In this study, we sprayed wood vinegar and butyrolactone on rapeseed and measured the morphological and physiological indicators, as well as the changes in transcriptome and metabolome. Combined with the comprehensive analysis of different levels of data, such as morphology, physiology, metabolism, and molecule data, we can further understand the regulation mechanisms of wood vinegar and one of its components, butyrolactone, on rapeseed growth, yield, quality, and low-temperature resistance.

## 2. Results and Analysis

### 2.1. Wood Vinegar and Butyrolactone Can Promote the Growth, Yield and Quality of Rapeseed

The effects of wood vinegar and butyrolactone treatment on rapeseed were studied by measuring the individual growth development at the seedling stage and the seed yield and quality at the maturity stage. Comparing the indicators of rapeseed at the seedling stage, it can be found (Table 1) that the indicators of rapeseed in M and D treatments are higher than those in the control. Among them, the leaf area and total dry weight of rapeseed treated with M increased by 8.67% and 8.77%, respectively, compared with the control. The root collar diameter, SPAD, leaf area, and total dry weight of rapeseed treated with D increased by 12.57%, 3.40%, 26.21%, and 36.19%, respectively, compared with the control, indicating that the application of wood vinegar mainly increased the biomass of rapeseed by increasing the leaf area, and butyrolactone mainly increased the biomass of rapeseed by increasing the root collar diameter, SPAD, and leaf area.

In terms of yield, Table 2 shows that the grain yield of M and D treatments increased by 15.53% and 26.91%, respectively, compared with the control. Comparing the yield components, it can be seen that the application of wood vinegar and butyrolactone significantly increased the number of effective pods of the plant, which increased by 10.44% and 17.29%, respectively, compared with the control. The results show that applying wood vinegar and butyrolactone could increase rapeseed yield mainly by increasing the number of effective pods per plant.

In terms of quality, Table 3 shows that the glucosinolate content of rapeseed treated with M and D decreased by 6.68% and 7.98%, respectively, compared with the control, but the effect on oil content, protein, linolenic acid, linoleic acid, and oleic acid of rapeseed was not significant compared with the control. The results show that applying wood vinegar and butyrolactone could improve the quality of rapeseed by reducing the content of glucosinolates.

### 2.2. Wood Vinegar and Butyrolactone Can Improve the Low-Temperature Tolerance of Rapeseed

The response of wood vinegar and butyrolactone to low temperature was further determined through sampling and analysis under low-temperature conditions (Table 4). The results show that the content of chlorophyll, proline, soluble sugar, and superoxide dismutase in M treatment increased by 39.73%, 40.54%, 39.12%, and 14.81%, respectively, compared with the control, and the difference reached a significant level. The contents of chlorophyll, soluble protein, and soluble sugar in D treatment increased by 21.92%, 14.27%, and 25.17%, respectively, compared with the control. In comparison, the contents of malondialdehyde and superoxide dismutase decreased by 53.33% and 46.34%, respectively. The results show that wood vinegar could effectively eliminate the free radicals produced in plants under low-temperature stress and reduce the damage caused by low temperatures. Butyrolactone treatment has stronger stress resistance at low temperatures by reducing the production of the harmful substance malondialdehyde.

### 2.3. Transcriptome Analysis of Rapeseed Treated with Wood Vinegar and Butyrolactone

To detect the difference in transcriptome between canola treated with wood vinegar and butyrolactone, we used the Illumina HiSeq 4000 platform for sequencing. Filter the low-quality readings with connectors and read lengths less than 80 bp in the original sequencing sequence, and obtain 246.90 Gb of pure data from 36 libraries (Appendix A). In our research, we mapped all high-quality data to the Brassica napus “ZS11” genome and finally located more than 89.87% of the pure data map on the reference genome (Appendix A). Under the same treatment, the correlation coefficients between biological repeats are greater than 0.95, indicating that the quality of RNA sequencing data is good (Appendix A).

We determined the expression levels of rapeseed leaves at 6 h, 12 h, 24 h, and 72 h, and conducted differential expression analysis. According to fold_change > 1.5 and mad > 0.5 and mean > 2, all expressed genes were divided into three sub-patterns: continuously up-regulated class, continuously down-regulated class and fluctuated class. The statistical results of differentially expressed genes among different comparison groups in this project are shown in Appendix A. Compared with the control, there were 681 differential genes in M treatment, of which 419 were up-regulated and 262 were down-regulated. There are 700 differential genes in treatment D, of which 267 are up-regulated and 433 are down-regulated (Figure 1).

The biological process, cell components, and molecular functions of differentially expressed genes treated with wood vinegar and butyrolactone were classified using GO analysis (Appendix A). In all differentially expressed gene comparisons, “metabolic process” and “cellular process”, “cellular part” and “cell”, as well as “catalytic activity” and “binding” are dominant in “biological process”, “cellular component”, and “molecular function” respectively.

Differentially expressed genes identified in rapeseed leaves treated with wood vinegar and butyrolactone were used for a KEGG pathway analysis to explore their potential functions. In total, 65 and 47 significant enrichment pathways were detected in the up-regulated and down-regulated genes in rapeseed after wood vinegar treatment (Figure 2 and Appendix A). The up-regulated genes in rapeseed are mainly involved in SNARE interactions in vesicular transport, riboflavin metabolism, phagosomes, non homologous terminal junctions, and diurnal rhythm plants. On the contrary, the down-regulated genes in rapeseed are related to tryptophan metabolism, other glycan degradation, lysine biosynthesis, and brassinosteroid biosynthesis. There are 50 and 71 significantly up-regulated and down-regulated gene enrichment pathways in rapeseed after butyrolactone treatment (Figure 2 and Appendix A). The up-regulated genes in rapeseed are mainly involved in thiamine metabolism, sulfur transmission system, sulfur metabolism, and glucosinolate biosynthesis. On the contrary, the down-regulated genes in rapeseed are involved in the biosynthesis of zeatin, the synthesis and degradation of ketones, the metabolism of nicotinate and nicotinamide, and the biosynthesis of brassinosteroid.

### 2.4. qPCR Validation of Transcriptome Data

In total, 27 genes were selected at random, and the specificity of primers was determined. Through a linear correlation analysis between differential expression multiples of transcriptome and qRT-PCR results (Figure 3), the correlation coefficient is R^2^ = 0.8378, which shows that the transcriptome sequencing results are accurate.

### 2.5. Identification and Analysis of Differential Metabolites

To detect the response of metabolites to wood vinegar and butyrolactone, LC-MS was used to analyze the metabolites in rapeseed leaves. We used all samples for principal component analysis, and found that all QC samples were closely clustered together (Figure 4). The dispersion between processed samples shows that the metabolic analyzer has a stable and reliable data detection function, which can be used for subsequent analysis. A partial least squares discriminant analysis (PLS-DA) and an orthogonal partial least squares discriminant analysis (OPLS-DA) (Appendix A) were performed on the metabolic spectra of wood vinegar and butyrolactone, respectively.

There are 76 differential metabolites in M treatment (Appendix A and Figure 5), and the ten metabolic pathways with the highest enrichment are phenylpropanoid biosynthesis, citrate cycle, biosynthesis of amino acids, alanine, aspartate and glutamate metabolism, and 2-Oxocarboxylic acid metabolism. There are 90 differential metabolites in treatment D (Appendix A and Figure 5), and the five metabolic pathways with the highest concentration are phenylpropanoid biosynthesis, pantothenate and CoA biosynthesis, nicotinate and nicotinamide metabolism, cyanoamino acid metabolism, and aminoacyl tRNA biosynthesis.

### 2.6. Differential Gene and Differential Metabolite Pathway Analysis

In order to analyze the relationship between differential genes and differential metabolites in rapeseed leaves under M and D treatments in detail, mapping integration association analysis was carried out on differential genes and differential metabolites between different treatments in KEGG pathway data, and a correlation network diagram was constructed. The results are as shown in Table 5. According to this Table, there are 35 differential genes and 11 differential metabolites mapped to 23 metabolic pathways between M treatment and non-treatment. The metabolic pathways in which the differential genes are enriched mainly include the biosynthesis of amino acids, carbon metabolism, and pentose and glucuronate interconversion. The differential metabolites are mainly mapped to phenylpropanoid biosynthesis, 2-oxocarboxylic acid metabolism, and other metabolic pathways. Interestingly, we found that the amino acid biosynthesis pathway and proline metabolism pathway were significantly affected, and carried out detailed network mapping (Figure 6). In the amino acid biosynthesis pathway, we identified seven genes and three metabolites. In addition, we identified the key genes *BnaA10G0179400ZS* and *BnaC02G0073500ZS* in this pathway, and the key metabolites L-phenylalanine and citric acid. In the proline metabolic pathway, we identified three genes and one metabolite. In addition, the key gene *BnaA10G0179400ZS* and the key metabolite spermidine in this pathway were identified. These results indicate that the differential regulation genes in the amino acid biosynthesis and proline metabolism pathway are highly related to the corresponding metabolites, indicating their importance for the biomass mediated by wood vinegar in rapeseed and the response to low-temperature stress.

There are 45 differential genes and 18 differential metabolites mapped to 24 metabolic pathways (Table 6) between D treatment and untreated treatment. The metabolic pathways enriched by differential genes mainly include phenylpropanoid biosynthesis, amino acid biosynthesis of amino acids, cysteine and methionine metabolism, purine metabolism, and carbon metabolism. The differential metabolite maps to the metabolic pathway mainly for phenylpropanoid biosynthesis, 2-oxoacetic acid metabolism, and glucosinolate biosynthesis. In addition, we found that the amino acid biosynthesis pathway and sphingolipid metabolism pathway were significantly affected, and carried out detailed network mapping (Figure 7). In the amino acid biosynthesis pathway, we identified six genes and three metabolites. In addition, the key gene *BnaC06G0249000ZS* and key metabolites L-aspartic acid, L-phenylalanine, and L-valine in this pathway were identified. In the sphingolipid metabolic pathway, we identified one gene and one metabolite. These results indicate that the differential regulation genes of amino acid biosynthesis and sphingolipid metabolism pathway are highly related to the corresponding metabolites, indicating their importance for butyrolactone-mediated biomass and low-temperature stress response in rapeseed.

## 3. Materials and Methods

### 3.1. Plant Materials and Growth Conditions

The test material was the hybrid rapeseed Huayouza 9, and was provided by Shengguang Seed Industry Co., Ltd. (Wuhan, China). The experiment was conducted at Huazhong Agricultural University (30°47′ N, 114°36′ E) from 2019 to 2020. The daily average temperature, solar radiation, and rainfall data were obtained from meteorological stations (AWS800, Campbell Scientific, Inc., Logan, UT, USA). The average daily temperature was 13.01 °C, the average daily solar radiation was 9.35 MJ/m^2^, and the total rainfall was 360.10 mm.

The bottom diameter, opening diameter, and height of the plastic pot for planting are 22.5 cm, 25.5 cm, and 27.0 cm, respectively. Each pot was filled with 14.0 kg silty loam soil. The basic chemical properties of the soil were ammonium nitrogen 1.53 mg/kg, nitrate nitrogen 7.89 mg/kg, available phosphorus 8.40 mg/kg, available potassium 112.88 mg/kg, organic matter 7.51 mg/kg, sulfur 170.00 mg/kg, and pH 6.00. Before sowing seed, 9.4 g of compound fertilizer (N15-P15-K15-S9) and 0.11 g boron fertilizer were applied to each pot as base fertilizer. Five seeds per pot were sown on 17 October 2019 and were thinned to two uniform seedlings at the three-leaf stage. In total, 3.0 g urea fertilizer was applied to each pot as top-dress fertilizer in the winter (4 December). These plants were managed using standard agricultural practices.

### 3.2. Spraying Treatment and Sampling

Wood vinegar came from poplar wood charcoal smoke and was provided by Hubei Chutian Biomass Energy Development Co., Ltd. (Wuhan, China). The components of wood vinegar are shown in Appendix A [14]. Butyrolactone is one of the main components of wood vinegar. Butyrolactone was provided by Sinopharm Chemical Reagent Co., Ltd. (Shanghai, China). Three treatments were set up in the experiment, namely, control (water), 400-fold diluted wood vinegar (M), and 2634 ug/mL butyrolactone (D), which were sprayed at the four-leaf stage of rapeseed with 5 mL per plant (19 November).

After spraying wood vinegar and butyrolactone for 6 h, 12 h, 24 h, and 72 h, the functional leaves from the three rapeseed plants were treated as experimental replicates. After sampling, they were quickly frozen into liquid nitrogen and stored in the refrigerator at −80 °C for transcriptome sequencing analysis. The samples were treated for 72 h for metabolomic analysis. From spraying to sampling 72 h after treatment, rapeseed was kept at a low temperature of 2–6 °C. After 72 h of treatment, the physiological indexes of leaves were measured. Samples of the transcriptome, metabolome, and physiological analysis were all three biological replicates.

Sampling was conducted at the seedling stage to investigate morphological indicators (3 December), and yield measurements were performed on 2 May 2020, with each treatment repeated ten times.

### 3.3. Determination of Morphology and Physiology

Five rapeseed plants were selected for analysis following fourteen days of spray treatment. Leaf area was quantified through an image analysis using ImageJ software (version 1.53a for Microsoft Windows). Leaf thickness was measured precisely with a thickness meter (SYS-YHD-2, Saiyasi, China). To determine dry matter weight, the leaves were oven-dried until reaching a constant weight. The SPAD value of the fourth fully expanded functional leaf, counted from the top to the base of the rapeseed plant, was measured using a SPAD-502 chlorophyll meter (Minolta, Osaka, Japan). The net photosynthetic rate and stomatal conductance were measured using a Li Cor6400XT portable photosynthesis analyzer (Li Cor Co., Ltd., Lincoln, NE, USA) six times. The protein, chlorophyll content, malondialdehyde, proline, and soluble sugars were determined using Coomassie brilliant blue [31], the ethanol extraction method [32], thiobarbituric acid method [33], ninhydrin method [34], and anthrone colorimetric [35], respectively. Superoxide dismutase (SOD), peroxidase (POD), and catalase (CAT) were measured using the WST-1 method, colorimetric method, and ultraviolet-visible spectrophotometry [36], respectively. The reactants were provided by Nanjing Jiancheng Biotechnology Co., Ltd. (Nanjing, China).

They were determined using Coomassie brilliant blue with bovine serum albumin as the standard [31]. Chlorophyll content was measured using the 95% ethanol extraction method [32]. Malondialdehyde was determined using the thiobarbituric acid method [33]. Proline was measured using the ninhydrin method [34]. Soluble sugar was determined using anthrone colorimetry [35]. Superoxide dismutase (SOD), peroxidase (POD), and catalase (CAT) were measured using the WST-1 method, colorimetric method, and ultraviolet-visible spectrophotometry [36], respectively. The reagents were provided by Nanjing Jiancheng Biotechnology Co., Ltd. (Nanjing, China).

### 3.4. Determination of Yield and Quality

Ten plants were sampled from each treatment at harvest time, and the plant height, root collar diameter, number of effective branches, number of effective pods, number of seeds per pod, 1000-seed weight, and biomass were determined.

A near-infrared analyzer (NIRSystem3750, Stockholm, Sweden) was used to determine the rapeseed’s quality.

### 3.5. RNA Isolation and Library Construction

To evaluate the transcriptome changes in rapeseed treated with wood vinegar and butyrolactone, we sequenced the RNA of leaf samples treated with 6 h, 12 h, 24 h, and 72 h, respectively, and the control group was treated with water spraying. We used the TRIzol kit (Novozan Biotechnology Co., Ltd., Nanjing, China) to extract total RNA, and used a spectrophotometer (NanoDrop 2000; Thermo Scientific, Waltham, MA, USA) to measure the concentration. An Agilent Bioanalyzer 2100 system was used to evaluate the RNA integrity. In total, 1.5 μg RNA samples were used for library construction, and RNA sequencing was performed on the Illumina HiSeq 4000 platform (Illumina, CA, USA).

### 3.6. RNA Sequencing Data Analysis

We used Fast QC to perform a quality control analysis on the original sequencing data, and used the low-quality sequence of the trimmatic filter belt connector to obtain pure data [37]. Mapping pure data to reference genome *Brassica napus* ZS11 (http://www.genoscope.cns.fr (accessed on 10 January 2021)), the gene expression was calculated using hisat2 and exons per thousand base fragments (FPKM) per million localization fragments and quantified using feature counts to display the gene expression level [38,39]. We conducted a correlation analysis and principal component analysis (PCA) on gene expression level. Screening differentially expressed genes according to the set threshold fold_change > 1.5 and mad > 0.5 and mean > 2, we used the DESeq2 software package (v1.22.2) to test the hypothesis of differentially expressed genes [40]. Phyper in R language was used to complete a GO enrichment analysis of the differentially expressed genes and KEGG enrichment analysis [41].

### 3.7. qRT PCR Analysis

To further confirm the gene expression level, we randomly selected 27 genes for real-time quantitative PCR (qRT PCR) analysis. We used SuperScript III reverse transcriptase (Invitrogen, Carlsbad, CA, USA) to reverse transcribe cDNA from 2 mg total RNA. We used the ABI 7500 real-time PCR system (ABI, Foster City, CA, USA) to conduct qRT-PCR in a 15 mL reaction, and the pass 2^−ΔΔCt^ method was used to analyze the relative expression of target genes [42]. The *BnActin7* gene was used as a control, and the primers can be retrieved from qPrimerDB [43]. Appendix A shows all of the primers used in this study. Three technical replicates and three biological replicates were conducted.

### 3.8. Metabolite Extraction

In total, 100 mg of each 3 samples were weighed after being ground with liquid nitrogen, then 50 mg of powder was divided into 1.5 mL EP tubes, 200 μL pre-cooled methanol/water (3:1, *v*/*v*) was added, vortex mixing was performed overnight at 4 °C at 13,000 rpm, centrifugation was performed at 4 °C for 15 min, the supernatant was taken and passed 0.22 μm after membrane filtration, and then was dried with nitrogen and stored at −80 °C. Before LC-MS mass spectrometry detection, we used 50 μL isopropanol/methanol/water (1:1:2, *v*/*v*/*v*) after redissolution and centrifugated the supernatant for detection.

### 3.9. LC-MS Analysis

Place the sample in the 4 °C automatic sampler and use the DIONEX UltiMate-3000 ultra high-performance liquid chromatography system (UHPLC) to separate the sample with the C18 chromatographic column [44]. In this, the injection volume was 3 μL. Column temperature was 45 °C and flow rate was 0.35 mL/min. Chromatographic mobile phase: 0.1% formic acid water (A), B: 0.1% formic acid acetonitrile (B). The chromatographic gradient elution procedure was as follows: 0–0.5 min, 98% A; 0.5–15 min, A changes linearly from 98% to 2%; 15–17 min, A maintained at 2%; 17.1–20 min, A changes linearly from 2% to 98%. Each sample was detected in positive and negative ion modes using electrospray ionization (ESI). In cation mode, the capillary voltage was set to 3500 V and the ion source temperature was kept at 320 °C. The samples were separated using UPLC and then analyzed using mass spectrometry with a Q-Exactive mass spectrometer (Thermo Fisher, Shanghai, China).

### 3.10. Metabolite Data Analysis

Peak alignment, retention time correction, and peak area extraction of raw data were performed using the Compound Discoverer 3.0 program. Metabolite structure identification adopts the method of precise mass number matching (<25 ppm) and secondary spectrogram matching to retrieve the MZcloud database and Chemspider database; apply the software SIMCA-P 14.1 (Umetrics, Umea, Sweden) for pattern recognition and conduct a multi-dimensional statistical analysis, including unsupervised principal component analysis (PCA) analysis, after the data are preprocessed using Pareto scaling, a Supervised partial least squares discriminant analysis (PLS-DA), and an orthogonal partial least squares discriminant analysis (OPLS-DA) [45]. The thresholds used to determine significant differences are VIP ≥ 1 and *p* < 0.05.

### 3.11. Association Analysis of Transcriptome and Metabolome

The differential genes and metabolites were mapped to the KEGG pathway database to obtain common pathway information. In addition, the information on differential genes in the transcriptome and differential metabolites in the metabolome was calculated using the Spearman algorithm to obtain the correlation coefficient (Corr) and correlation *p* value. According to the absolute value of correlation coefficient |Corr| > 0.85 and correlation *p* < 0.05, differential expression genes and differential metabolites were screened. The correlation network diagram was constructed using the software Cytoscape v3.3.0 [46].

### 3.12. Data Processing and Analysis

Statistix 9.0 software (Analytical Software, Tallahassee, FL, USA) was used for ANOVA of data, and R 4.0.1 language was used for drawing (http://www.r-project.org (accessed on 10 January 2021)). The least significant difference (LSD) method based on a 0.05 level was used for the different analyses among the treatments.

## 4. Discussion

### 4.1. Wood Vinegar and Butyrolactone Can Improve the Yield of Rapeseed

Our comparative approach revealed that butyrolactone, when applied separately, produced similar effects to those of wood vinegar in enhancing cold tolerance. This suggests that butyrolactone is a key active ingredient in wood vinegar. Previous research shows that applying wood vinegar can promote the accumulation of dry matter in rice and cotton [47,48]. Soaking seeds with wood vinegar can increase the root surface area of wheat seedlings under drought stress and promote the growth and development of flue-cured tobacco and dry matter accumulation [11]. This study demonstrates that low-temperature stress significantly reduces the leaf area of rapeseed, while spraying wood vinegar promoted leaf area expansion and dry matter accumulation under these conditions. Wood vinegar effectively alleviated the inhibitory effects of low-temperature stress, suggesting its potential to enhance rapeseed growth in cold environments. This may be attributed to the fact that the alcohols, phenols, and ketones contained in wood vinegar regulate plant growth and development and improve nutrient absorption capacity. In addition, the trace elements contained in wood vinegar are also absorbed and utilized by plants [49].

### 4.2. Physiological Changes in Rapeseed in Response to Low-Temperature Stress Enhanced by Wood Vinegar and Butyrolactone

The damage of low-temperature stress to plants is manifested in many aspects. First, the plant produces relevant physiological and biochemical reactions, which then affect plant growth and morphogenesis, leading to a reduction in leaf area and a significant decline in biomass [50]. Malondialdehyde is an important indicator of plant stress resistance strength [51]. Research shows that under low-temperature stress, the malondialdehyde content of plants rises sharply, and the osmotic adjustment ability decreases [52]. This study shows that wood vinegar and butyrolactone treatment can effectively reduce the content of malondialdehyde in rapeseed under low-temperature stress, alleviate the damage in the cell membrane of rapeseed, reduce the accumulation of active oxygen and chlorophyll decomposition, and to some extent enhance the cold resistance of rapeseed. The osmoregulation substances in plants are closely related to cold resistance [53,54]. The results show that spraying appropriate concentrations of wood vinegar and butyrolactone could increase the content of proline, soluble sugar, and soluble protein in rapeseed seedlings under low-temperature stress, improve the water holding capacity of cells, prevent cell dehydration, and ensure the normal physiological and biochemical processes of cells. In addition, the increase in proline content improves the solubility of proteins in water [55], maintains the conformation of various enzymes in plants, and enhances the cold resistance of rapeseed seedlings. 

Our results show that applying wood vinegar can improve the SOD activity of rapeseed under low temperatures. The reason may be that wood vinegar contains phenols, organic acids, and other substances that can affect the physiological activity of plants. On the one hand, the phenols in wood vinegar have good antioxidant and free radical scavenging capacity, improve cell membrane stability, and reduce the impact of low temperature on the proportion and activity of membrane-bound enzymes [56]. On the other hand, studies have shown that wood vinegar can stimulate the defense mechanisms of rapeseed seedlings through acid stimulation and promote an increase in antioxidant enzyme activity of seedlings. Therefore, treating rapeseed seedlings with wood vinegar can effectively eliminate superoxide anion radicals and H_2_O_2_ produced by stress through SOD to adapt to its stress. 

### 4.3. Transcriptome Analysis of Wood Vinegar and Butyrolactone on Rapeseed

The regulation of wood vinegar and butyrolactone on rapeseed growth and low-temperature stress is related to plant hormone response. Receptor-like protein kinase (RLK-LRR) with rich leucine repeats is a major and very conservative plant kinase family [57]. LRR-RLK plays an important role in plant cell growth, signal transduction, and stress defense. Previous studies have shown that LRR-RLK mediates leaf cell senescence through its synergistic function with ethylene and auxin [58]. Both auxin and ethylene are involved in controlling the grain filling of crops under low temperature [59]. The filling of grains in crops is closely related to the senescence products of the whole plant. It depends on the direct transfer of current carbohydrates to grains and requires the redistribution of retained assimilates in vegetative organs [60]. In this study, wood vinegar and butyrolactone showed a relationship with plant hormone response in low-temperature treatment. These genes include the auxin response genes *BnaC06G0249000ZS* and *BnaA10G0179400ZS* [61,62]. By promoting the function of plant auxin, it can restore plant growth caused by low-temperature stress and low-temperature and delayed senescence. On the other hand, under low-temperature stress, wood vinegar and butyrolactone may have a greater ability to preserve more assimilates for redistribution. Redistributing assimilates to grains and enhancing auxin transport and ethylene signal transduction lead to early senescence after low-temperature stress. Therefore, wood vinegar and butyrolactone may activate the auxin/ethylene signaling pathway. A recent study confirmed that the melatonin response in rice is closely related to the auxin signaling pathway [63]. In addition, ABA may be a downstream signal of melatonin in abiotic stress response [64]. Therefore, the hormones in rapeseed treated with wood vinegar and butyrolactone can act synergistically as the key regulators of the plant stress response.

### 4.4. Metabolomic Analysis of Wood Vinegar and Butyrolactone on Rapeseed

Metabolites directly reflect the impact of environmental changes on plants [65]. Our comparative metabolite analysis showed that wood vinegar and butyrolactone induced significant changes in metabolite content in rapeseed seedlings. The abundance of several metabolites, including (L-glutathione oxidation, kaempferol) has been greatly increased. These metabolites were previously reported to have antioxidant effects [66,67], reflecting their internal role in the response to environmental pressure mediated by wood vinegar and butyrolactone. The results show that wood vinegar and butyrolactone activated many antioxidant pathways to protect seedlings from low-temperature stress. Moreover, wood vinegar and butyrolactone can increase the content of soluble alcohols (plant sphingosine, naphthalene triol, decyl panthenol, propylene glycol), citrate and other compatible solutes under low temperatures. Physiologically, many of these metabolites may play an osmotic role in the cytoplasm under low-temperature stress [68]. Thus, the osmotic pressure of cells is increased, which is conducive to improving the low-temperature tolerance of plants. In addition, other metabolites, such as aspartic acid, L-phenylalanine, proline, valine, are highly accumulated in seedlings treated with wood vinegar and butyrolactone, indicating their beneficial physiological role in their response to low-temperature stress mediated by wood vinegar and butyrolactone. Previous studies have found that amino acids (such as proline and ornithine) are the “cheapest” forms of osmotic adaptation [69]. Fewer changes in amino acid metabolism and TCA cycle are related to better branch growth [70]. In this study, low-temperature stress significantly increased the amino acid biosynthesis (such as proline and asparagine) of all genotypes. The enhancement of amino acid metabolism and TCA cycle may be related to increased energy supply and biomass of osmotic synthesis under low temperatures. Alternatively, they can compensate for ion imbalance and promote nitrogen/carbon metabolism [71]. These results indicate that the accumulation of these metabolites induced by wood vinegar and butyrolactone may directly lead to an improvement in the low-temperature tolerance of seedlings.

### 4.5. Integrated Transcriptional and Metabolomic Analysis of KEGG Pathway of Wood Vinegar and Butyrolactone on Rapeseed

Recently, the potential mechanisms of plant metabolite changes and related genetic mechanisms have been studied [72,73,74]. As the final product of gene expression, metabolites are the direct relationship between genotype and phenotype, which suggests that data on metabolite profiles may be helpful to clarify the molecular basis of growth and low-temperature resistance mediated by wood vinegar and butyrolactone. In this study, the biosynthesis of amino acids in rapeseed treated with wood vinegar and butyrolactone was studied. Amino acid metabolic pathways related to nitrogen metabolism include cyano amino acid metabolism, phenylalanine metabolism, alanine, aspartic acid and glutamic acid metabolism, and tryptophan metabolism. In addition, we found that treatment significantly affected phenylpropane biosynthesis and carbon metabolism in rapeseed. The phenylpropane metabolic pathway is closely related to lignin synthesis [75], while the lignin biosynthesis pathway is related to the utilization of photosynthetic carbon [76]. Our results indicate that more photosynthetic carbon can be transferred to lignin synthesis through glycolysis or PPP after treatment. Therefore, wood vinegar and butyrolactone may affect the C/N balance of rapeseed by regulating the distribution of photosynthetic carbon, the assimilation of nitrogen and aromatic amino acids, and the biosynthesis of phenylpropane, and then affect the yield and photosynthesis of rapeseed.

## 5. Conclusions

Research has shown that the foliar application of wood vinegar and its active ingredient butyrolactone significantly promotes the growth of rapeseed and enhances plant resistance at low temperatures. Overall, butyrolactone has a greater promoting effect on biomass and yield at low temperatures. The wood vinegar containing hundreds of organic compounds may have relatively reduced differential gene expression due to the interactions of multiple substances. Therefore, compared with the control group, the transcriptome treated with wood vinegar and butyrolactone had 681 and 700 differentially expressed genes, respectively, while the metabolome had 76 and 90 differentially expressed metabolites. The transcriptome and metabolome analyses and their correlations showed that both wood vinegar and butyrolactone can enhance auxin transport and ethylene signal transduction. The key genes of auxin, *BnaC06G0249000ZS* and *BnaA10G0179400ZS*, as well as the key metabolite L-phenylalanine, were identified. In terms of physiological indicators, both can effectively reduce the content of malondialdehyde in plants under low-temperature stress, increase the content of proline, soluble sugar, and soluble protein in individuals, and effectively reduce the production of superoxide anion radicals and H_2_O_2_. The difference is that wood vinegar has a strong expression of key genes *BnaA10G0179400ZS* and *BnaC02G0073500ZS* in the amino acid biosynthesis pathway, as well as the key metabolite citric acid. Wood vinegar primarily regulates rapeseed osmotic stress through arginine and proline metabolism and enhances antioxidant defenses via flavonoid biosynthesis and glutathione metabolism. In contrast, butyrolactone, through the key gene *BnaC06G0249000ZS* in amino acid biosynthesis, strongly influences metabolites like L-aspartic acid and L-valine, which are involved in osmotic regulation and managing malondialdehyde accumulation through sphingolipid metabolism and the biosynthesis of keratin, amber, wax, and fatty acids.

## Figures and Tables

**Figure 1 ijms-25-09757-f001:**
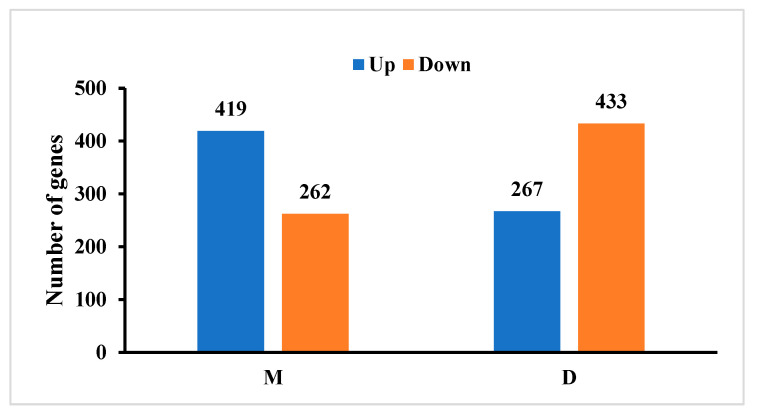
Number of differentially expressed genes. M, Wood vinegar. D, Butyrolactone.

**Figure 2 ijms-25-09757-f002:**
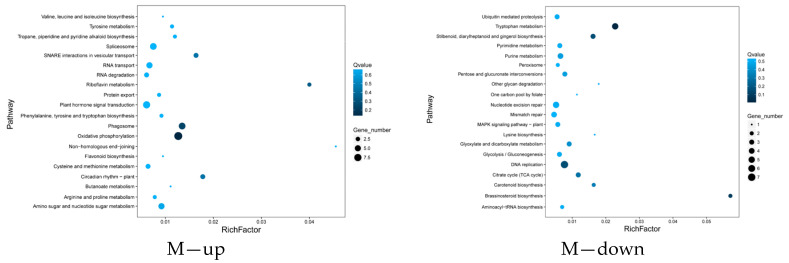
Differential gene KEGG pathway. M, Wood vinegar. D, Butyrolactone.

**Figure 3 ijms-25-09757-f003:**
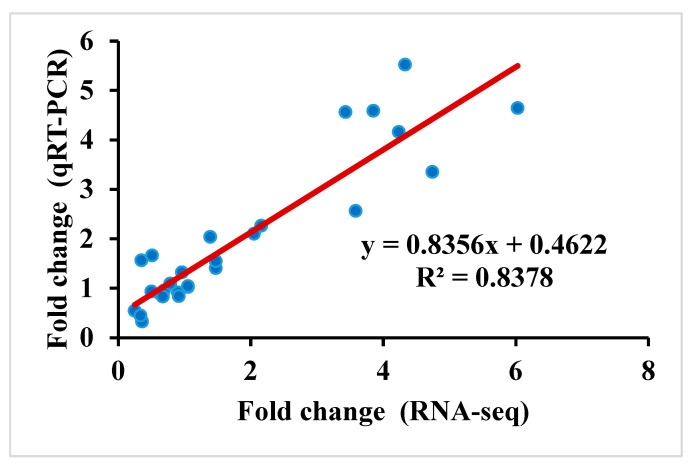
Correlation analysis between transcriptome and qRT-PCR results. Each blue dot represents an individual gene, where its position reflects the relationship between RNA-seg data and gRT-PCR results for that gene. The red line indicates the best-fit linear regression model.

**Figure 4 ijms-25-09757-f004:**
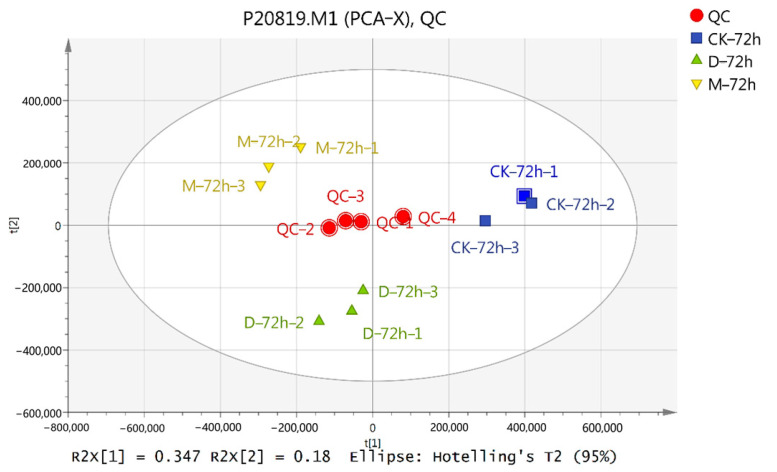
Principal component analysis between samples. QC, Quality control. M, Wood vinegar. D, Butyrolactone.

**Figure 5 ijms-25-09757-f005:**
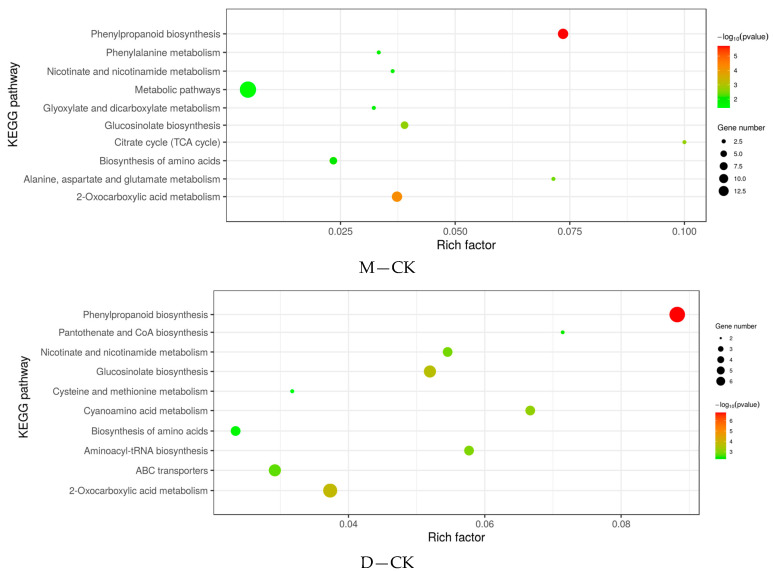
Differential metabolite KEGG pathway. CK, Control. M, Wood vinegar. D, Butyrolactone.

**Figure 6 ijms-25-09757-f006:**
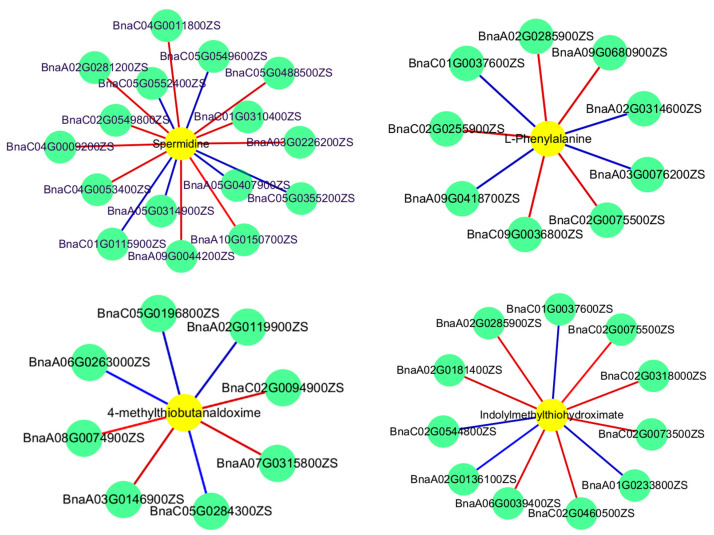
Correlation analysis of differentially expressed genes and metabolites in M treatment. Gene–metabolite pairs are connected within the network by edges. The green and yellow circles represent genes and metabolites, respectively, the red line represents a positive correlation, and the blue line represents a negative correlation.

**Figure 7 ijms-25-09757-f007:**
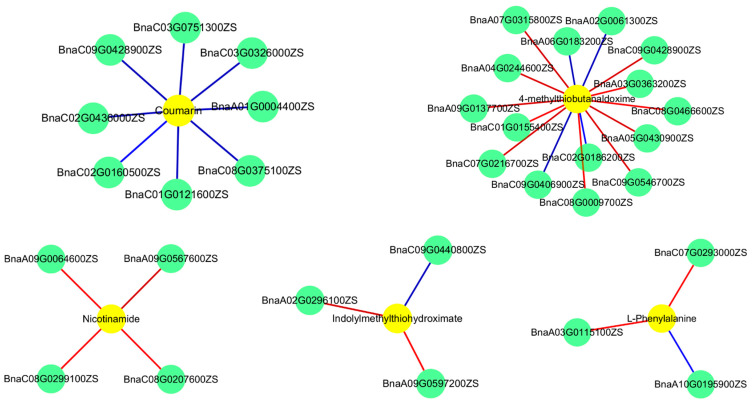
Correlation analysis of differentially expressed genes and metabolites in D treatment. Gene–metabolite pairs are connected within the network by edges. The green and yellow circles represent genes and metabolites, respectively, the red line represents a positive correlation, and the blue line represents a negative correlation.

**Table 1 ijms-25-09757-t001:** The effects of wood vinegar and butyrolactone treatments on the individual development of rapeseed seedlings.

Treatment	Root Collar Diameter (mm)	Number of Total Leaves	Number of Green Leaves	Root Dry Weight (g/plant)	Leaf Dry Weight (g/plant)	Total Dry Weight (g/plant)	SPAD	Leaf Area (cm^2^)	Leaf Thickness (mm)	Net Photosynthetic Rate (µmol m⁻^2^ s⁻^1^)	Stomatal Conductance (µmol m⁻^2^ s⁻^1^)
CK	7.16 b	6.50 a	6.33 a	1.38 b	12.41 b	13.79 b	43.47 b	461.38 c	0.517 a	29.67 a	0.91 a
M	7.65 ab	6.50 a	6.33 a	1.48 ab	13.52 b	15.00 b	44.90 a	501.38 b	0.518 a	29.84 a	0.93 a
D	8.06 a	6.83 a	6.67 a	1.92 a	16.86 a	18.78 a	44.95 a	582.31 a	0.519 a	30.78 a	0.97 a

Note: Statistix 9.0 software (Analytical Software, Tallahassee, FL, USA) was used for ANOVA of data. According to the LSD test, the lowercase letters after the numbers in each column of this Table indicate significant differences (*p* < 0.05), (*n* = 6). CK, Control. M, Wood vinegar. D, Butyrolactone.

**Table 2 ijms-25-09757-t002:** Effects of wood vinegar and butyrolactone treatment on rapeseed yield and composition factors.

Treatment	Plant Height (cm)	Root Collar Diameter (mm)	Stem Diameter (cm)	Branch Position (cm)	Length of Main Inflorescence (cm)	Branch Number per Plant	Pods Number on the Main Stem	Branch Pods Number per Plant	Pods Number per Plant	Seeds Number per Pod	Seed Weight (g/1000)	Yield per Plant (g)	Dry Weight above Ground per Plant (g)
CK	132.95 b	1.64 a	1.36 a	30.70 b	58.86 b	8.14 a	66.83 c	183.83 a	250.67 b	25.97 a	2.61 a	16.94 b	54.86 b
M	139.00 a	1.66 a	1.44 a	36.30 a	62.39 b	8.00 a	73.67 b	203.17 a	276.83 ab	27.47 a	2.61 a	19.57 a	62.90 a
D	140.90 a	1.65 a	1.44 a	32.89 ab	69.87 a	8.00 a	81.33 a	212.67 a	294.00 a	28.34 a	2.62 a	21.50 a	67.48 a

Note: Statistix 9.0 software (Analytical Software, Tallahassee, FL, USA) was used for ANOVA of data. According to the LSD test, the lowercase letters after the numbers in each column of this Table indicate significant differences (*p* < 0.05), (*n* = 6). CK, Control. M, Wood vinegar. D, Butyrolactone.

**Table 3 ijms-25-09757-t003:** Effects of wood vinegar and butyrolactone treatment on rapeseed quality.

Treatment	Protein Content (%)	Oil Content (%)	Glucosinolate (μmol/g)	Linolenic Acid (%)	Linoleic Acid (%)	Oleic Acid (%)
CK	16.18 a	46.91 a	33.22 a	7.03 a	16.26 a	69.49 a
M	15.87 a	47.22 a	31.00 b	6.98 a	16.26 a	69.93 a
D	15.59 a	47.54 a	30.57 b	6.66 a	16.91 a	69.60 a

Note: Statistix 9.0 software (Analytical Software, Tallahassee, FL, USA) was used for ANOVA of data. According to the LSD test, the lowercase letters after the numbers in each column of this Table indicate significant differences (*p* < 0.05), (*n* = 6). CK, Control. M, Wood vinegar. D, Butyrolactone.

**Table 4 ijms-25-09757-t004:** The effects of different treatments on the physiological indexes of rapeseed under low temperature.

3.	Chlorophyll Content (mg/g)	Soluble Protein (g prot/L)	Proline (μg/g)	Malondialdehyde (μmol/g)	Soluble Sugar (mg/g)	SOD (U/mg prot)	POD (U/mg prot)	CAT (U/mg prot)
CK	0.73 b	10.65 b	69.90 b	0.30 a	12.91 b	13.10 b	8.00 a	7.58 a
M	1.02 a	11.16 ab	98.24 a	0.22 ab	17.96 a	15.04 a	7.00 a	7.54 a
D	0.89 a	12.17 a	88.09 ab	0.14 b	16.16 a	7.03 c	6.33 a	6.99 a

Note: According to the LSD test, the lowercase letters after the numbers in each column of this Table indicate significant differences (*p* < 0.05), (*n* = 6). CK, Control. M, Wood vinegar. D, Butyrolactone. SOD, superoxide dismutase. POD, peroxidase. CAT, catalase.

**Table 5 ijms-25-09757-t005:** Differential genes and differential metabolite pathways of wood vinegar treatment.

KEGG Pathway	Genes	Metabolite
Phenylpropanoid biosynthesis	BnaC01G0012800ZS, novel.18876, BnaA10G0034100ZS	L-Phenylalanine, Spermidine, Sinapate, Ferulate, trans-2-Hydroxycinnamate
2-Oxocarboxylic acid metabolism	BnaC02G0073500ZS, BnaA10G0179400ZS, BnaA04G0241000ZS	Indolylmethylthiohydroximate, 2-Oxoglutarate, L-Phenylalanine, Citrate, 4-Methylthiobutanaldoxime
Citrate cycle (TCA cycle)	BnaA06G0006700ZS, BnaA08G0287200ZS, BnaC04G0228700ZS, BnaC02G0554900ZS	2-Oxoglutarate, Citrate
Alanine, aspartate, and glutamate metabolism	BnaA10G0179400ZS	2-Oxoglutarate, Citrate
Biosynthesis of amino acids	BnaA10G0179400ZS, BnaA03G0384700ZS, BnaC04G0009200ZS, BnaC02G0073500ZS, BnaA04G0241000ZS, BnaC04G0313900ZS, BnaC09G0538200ZS	2-Oxoglutarate, L-Phenylalanine, Citrate
Phenylalanine metabolism	BnaA10G0179400ZS	L-Phenylalanine, trans-2-Hydroxycinnamate
Glyoxylate and dicarboxylate metabolism	BnaC01G0228200ZS, BnaC01G0037600ZS, BnaA01G0032600ZS	2-Oxoglutarate, Citrate
Carbon metabolism	BnaA10G0179400ZS, BnaC09G0057900ZS, BnaC02G0554900ZS, BnaC01G0037600ZS, BnaA01G0032600ZS, BnaA06G0006700ZS, BnaC04G0228700ZS	2-Oxoglutarate, Citrate
Arginine biosynthesis	BnaA10G0179400ZS, BnaA04G0241000ZS	2-Oxoglutarate
C5-Branched dibasic acid metabolism	BnaC02G0073500ZS	2-Oxoglutarate
Phenylalanine, tyrosine, and tryptophan biosynthesis	BnaA10G0179400ZS, BnaA03G0384700ZS	L-Phenylalanine
Lysine biosynthesis	BnaC09G0538200ZS	2-Oxoglutarate
Glutathione metabolism	BnaA09G0015300ZS	Spermidine
Butanoate metabolism	BnaC02G0073500ZS	2-Oxoglutarate
Cyanoamino acid metabolism	novel.18876	L-Phenylalanine
Ascorbate and aldarate metabolism	BnaC02G0069000ZS	2-Oxoglutarate
Aminoacyl-tRNA biosynthesis	BnaC08G0411300ZS, BnaA09G0059500ZS, BnaA04G0128500ZS	L-Phenylalanine
Pentose and glucuronate interconversions	BnaA01G0353900ZS, BnaA07G0072600ZS, BnaC01G0473100ZS, Bnascaffold0027G0053200ZS, BnaC02G0069000ZS, BnaA05G0485000ZS	2-Oxoglutarate
Tropane, piperidine and pyridine alkaloid biosynthesis	BnaC03G0271400ZS, BnaA10G0179400ZS	L-Phenylalanine
Flavonoid biosynthesis	BnaC02G0537300ZS	Kaempferol
Tyrosine metabolism	BnaA04G0155600ZS, BnaA10G0179400ZS	2-Hydroxy-3-(4-hydroxyphenyl)propenoate
Arginine and proline metabolism	BnaA10G0179400ZS, BnaC09G0569400ZS, BnaC04G0313900ZS	Spermidine
Tryptophan metabolism	novel.24812, BnaA01G0142600ZS, novel.24814, BnaA01G0032600ZS, BnaC01G0037600ZS	Indolylmethylthiohydroximate

**Table 6 ijms-25-09757-t006:** Differential genes and differential metabolite pathways of butyrolactone treatment.

KEGG Pathway	Genes	Metabolite
Phenylpropanoid biosynthesis	BnaA09G0566300ZS, BnaC04G0553100ZS, BnaC03G0021400ZS, BnaC06G0428100ZS, BnaC06G0181900ZS, BnaA04G0299600ZS	L-Phenylalanine, Sinapate, 4-Coumarate, Ferulate, 5-Hydroxyferulic acid methyl este, Coumarin
2-Oxocarboxylic acid metabolism	BnaA08G0059700ZS, BnaA01G0008700ZS, BnaA04G0241000ZS	L-Aspartate, L-Phenylalanine, L-Valine, Indolylmethylthiohydroximate, 4-Methylthiobutanaldoxime
Glucosinolate biosynthesis	BnaA01G0008700ZS, BnaA08G0059700ZS	L-Phenylalanine, L-Valine, Indolylmethylthiohydroximate, 4-Methylthiobutanaldoxime
Cyanoamino acid metabolism	BnaA09G0566300ZS, BnaA04G0299600ZS, BnaC03G0021400ZS	L-Aspartate, L-Phenylalanine, L-Valine
Aminoacyl-tRNA biosynthesis	novel.636	L-Aspartate, L-Phenylalanine, L-Valine
Nicotinate and nicotinamide metabolism	BnaA02G0083500ZS, BnaC02G0270800ZS	L-Aspartate, Nicotinamide, N-Methylnicotinate
Biosynthesis of amino acids	BnaC04G0009200ZS, BnaC06G0249000ZS, BnaA08G0001500ZS, BnaC03G0816400ZS, BnaA04G0241000ZS, BnaC02G0319900ZS	L-Aspartate, L-Phenylalanine, L-Valine
Cysteine and methionine metabolism	BnaA03G0377000ZS, BnaC03G0461700ZS, BnaC03G0816400ZS, BnaA08G0001500ZS, BnaC04G0009200ZS, novel.50243, BnaC07G0293000ZS	L-Aspartate, Dehydroalanine
Carbon fixation in photosynthetic organisms	BnaC06G0249000ZS, BnaA03G0427900ZS, BnaC02G0319900ZS, BnaC02G0209200ZS, BnaC09G0325700ZS	L-Aspartate
Arginine biosynthesis	BnaA04G0241000ZS	L-Aspartate
Sphingolipid metabolism	BnaA04G0244600ZS	Phytosphingosine
Cutin, suberine, and wax biosynthesis	novel.25263	Hexadecanoic acid
Beta-Alanine metabolism	BnaC08G0024500ZS	L-Aspartate
Valine, leucine, and isoleucine degradation	BnaC07G0327100ZS, BnaC08G0024500ZS, BnaC02G0488700ZS	L-Valine
Fatty acid degradation	BnaA05G0470300ZS, BnaA05G0181300ZS, BnaC07G0327100ZS, BnaA09G0597200ZS, BnaC02G0488700ZS	Hexadecanoic acid
Fatty acid biosynthesis	BnaA05G0470300ZS, BnaA05G0181300ZS, BnaA02G0298900ZS	Hexadecanoic acid
Phenylalanine metabolism	BnaC04G0553100ZS, BnaC06G0181900ZS	L-Phenylalanine
Tropane, piperidine, and pyridine alkaloid biosynthesis	novel.8259, BnaA05G0141600ZS	L-Phenylalanine
Biosynthesis of unsaturated fatty acids	BnaC02G0488700ZS	Hexadecanoic acid
Tyrosine metabolism	BnaA09G0597200ZS	4-Coumarate
Tryptophan metabolism	BnaA08G0059700ZS, BnaA01G0008700ZS, BnaA01G0142600ZS, novel.8529, BnaC07G0327100ZS	Indolylmethylthiohydroximate
Purine metabolism	BnaC09G0106200ZS, BnaA01G0189300ZS, BnaA01G0356800ZS, BnaC02G0270800ZS, BnaC08G0375100ZS, BnaA09G0434400ZS, BnaC04G0298300ZS, BnaC02G0212200ZS	Adenosine
Carbon metabolism	novel.18384, BnaA03G0427900ZS, BnaC03G0816400ZS, BnaC06G0249000ZS, BnaA08G0001500ZS, BnaC09G0325700ZS, BnaC02G0209200ZS, BnaC02G0319900ZS, BnaC07G0327100ZS, BnaC08G0024500ZS	L-Aspartate
Fatty acid metabolism	BnaA05G0470300ZS, BnaC02G0488700ZS, BnaA05G0181300ZS, BnaC07G0327100ZS, BnaA02G0298900ZS	Hexadecanoic acid

## Data Availability

The authors can provide data upon formal request.

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
