# Peer review of "Analysis of the Mechanism of Wood Vinegar and Butyrolactone Promoting Rapeseed Growth and Improving Low-Temperature Stress Resistance Based on Transcriptome and Metabolomics"

_ijms, 2024, doi:10.3390/ijms25179757_

Round 1

Reviewer 1 Report

Comments and Suggestions for Authors

The paper is interesting as it makes an original approach to characterize the mechanism of action of a substance able to protect plants from abiotic stress.

The main problem I have found is that the discussion is too descriptive and repetitive with the results section and does not clearly answers the starting questions: is gamma-butyrolactone the responsible of the effect?  Which is the main mechanism that explains the stress tolerance triggered by wood vinegar? So focusing the discussion, or adding a conclusion with a clear answer for these two questions will greatly increase the value of the manuscript.

Other major point is that authors have validate the trascriptomic data by RT-PCR of random genes. I suggest to include a figure with RT PCR of several of the genes included in figures 6 and 7, as they are key ones that support the conclusions.

Other points:

Line 19: metabolomic.

Tables 1-3: include in the legend which statistical analysys have you performed.

Author Response

Dear Reviewer:

We thank the reviewer for their constructive comments and suggestions of this manuscript. We addressed each point in turn below. Any revisions have been clearly highlighted using the red fonts to make it easily visible to the reviewer. I would appreciate your further consideration of our manuscript. Please let me know if you have any further comments and suggestions.

Comments1:The main problem I have found is that the discussion is too descriptive and repetitive with the results section and does not clearly answers the starting questions: is gamma-butyrolactone the responsible of the effect?  Which is the main mechanism that explains the stress tolerance triggered by wood vinegar? So focusing the discussion, or adding a conclusion with a clear answer for these two questions will greatly increase the value of the manuscript.

Response1:Thank you very much for pointing out this. Research has shown that foliar application of wood vinegar and its active ingredient butyrolactone significantly promotes the growth of rapeseed and enhances plant resistance at low temperatures. Overall, butyrolactone has a greater promoting effect on biomass and yield at low temperatures. (Lines 63- 67). We added 5. Conclusion. The similarities and differences between wood vinegar and butyrolactone were discussed in detail. (Lines 611- 632)

Comments2:Other major point is that authors have validate the trascriptomic data by RT-PCR of random genes. I suggest to include a figure with RT PCR of several of the genes included in figures 6 and 7, as they are key ones that support the conclusions.

Response2:Many thanks for your suggestion. Using RT-PCR of random genes to validate transcriptomic data is a commonly used experimental method aimed at confirming the accuracy and reliability of transcriptome sequencing results. This method evaluates the credibility of the entire transcriptome data by selecting a subset of random genes for expression level detection. We conducted RT-PCR by randomly selecting genes to verify the accuracy of transcriptomic data. The results also demonstrate the reliability of transcriptomic data. Our subsequent research will focus on validating the gene functions of these key genes, providing a detailed explanation of the mechanisms by which these genes contribute to the growth and low temperature resistance of rapeseed, and publishing it in subsequent articles.

Comments3:Line 19: metabolomic.

Response3:Thank you very much for pointing out this. "metabonomics" corrected to " metabolomic ". (Lines 18)

Comments4:Tables 1-3: include in the legend which statistical analysys have you performed.

Response4:Many thanks for your suggestion. Added "Statistix 9.0 software (Analytical Software, Tallahassee, Florida, USA) was used for ANOVA of data." in the notes. (Lines 264, 277, 288)

Reviewer 2 Report

Comments and Suggestions for Authors

The manuscript, entitled: "Analysis of the mechanism of wood vinegar and butyrolactone promoting rapeseed growth and improving stress resistance based on transcriptome and metabonomics" present research which is aimed to increase the cold tolerance of canola.

The introduction is well detailed and presents related research well. The materials and methods chapter is sufficiently detailed and reproducible. It combines well plant physiological observations with transcriptome and metabonomic analyzes used to elucidate the molecular mechanism. At the same time, 2.3. chapter, I recommend a clear definition of the abbreviations SOD, POD, CAT, when they are first mentioned (manuscript line 157).

The presentation of the results is sufficiently detailed and informative. At the same time, a few formal amendments and corrections are necessary:
- In the case of table 1, the name of the CK is missing.
- In the case of figure 4, the definition of QC is missing.
- In the case of figure 5 and figure 6, the title has slipped in the manuscript. Precise positioning is required.

The discussion chapter interprets the results well and compares them with similar studies. Contains a sufficient number of references (76).

The tests were carried out in sufficient detail. The presentation of the results is suitable for the treatments with the wood vinegar and butyrolactone compounds that were carried out at the age of a few leaves. The effect on rapeseed growth and improving stress resistance was significantly different.

After implementing the suggested corrections, I recommend publishing the manuscript in the form of a scientific article.

Author Response

Dear Reviewer:

We thank the reviewer for their constructive comments and suggestions of this manuscript. We addressed each point in turn below. Any revisions have been clearly highlighted using the red fonts to make it easily visible to the reviewer. I would appreciate your further consideration of our manuscript. Please let me know if you have any further comments and suggestions.

Comments1: 2.3. chapter, I recommend a clear definition of the abbreviations SOD, POD, CAT, when they are first mentioned (manuscript line 157).

Response1:Thank you very much for pointing out this. SOD, POD and CAT have been corrected it to "superoxide dismutase (SOD), peroxidase (POD), and catalase (CAT)”. (Lines 164)

Comments2:In the case of table 1, the name of the CK is missing.

Response2:Many thanks for your suggestion. CK in the full text notes has been annotated as " CK, Control.". Line 266, etc)

Comments3:In the case of figure 4, the definition of QC is missing.

Response3:Thanks for your suggestion. Table 4 has added "QC, Quality control.". (Lines 388). QC samples (quality control samples) play an important role in metabolomics research, ensuring data stability, reproducibility, and accuracy. By monitoring the quality of the experimental process in real-time, QC samples help identify systematic errors and drifts in analysis, and provide references for data standardization and correction, thereby optimizing experimental conditions, reducing batch effects, and ensuring the reliability of the final data. This experiment has strong data stability, good repeatability, and high accuracy.

Comments4:In the case of figure 5 and figure 6, the title has slipped in the manuscript. Precise positioning is required.

Response4:Thank you very much for pointing out this. The title has been corrected. (Lines 407)

Round 2

Reviewer 1 Report

Comments and Suggestions for Authors

Paper has been significatively improved.

I can recommend publication.